# Arabinoxylans as Functional Food Ingredients: A Review

**DOI:** 10.3390/foods11071026

**Published:** 2022-04-01

**Authors:** Emanuele Zannini, Ángela Bravo Núñez, Aylin W. Sahin, Elke K. Arendt

**Affiliations:** 1School of Food and Nutritional Sciences, University College Cork, T12 K8AF Cork, Ireland; abravonunez@ucc.ie (Á.B.N.); aylin.sahin@ucc.ie (A.W.S.); e.arendt@ucc.ie (E.K.A.); 2APC Microbiome Ireland, University College Cork, T12 K8AF Cork, Ireland

**Keywords:** arabinoxylans, dietary fibre, health benefits, arabinoxylans food applications

## Abstract

The health benefits of fibre consumption are sound, but a more compressive understanding of the individual effects of different fibres is still needed. Arabinoxylan is a complex fibre that provides a wide range of health benefits strongly regulated by its chemical structure. Arabinoxylans can be found in various grains, such as wheat, barley, or corn. This review addresses the influence of the source of origin and extraction process on arabinoxylan structure. The health benefits related to short-chain fatty acid production, microbiota regulation, antioxidant capacity, and blood glucose response control are discussed and correlated to the arabinoxylan’s structure. However, most studies do not investigate the effect of AX as a pure ingredient on food systems, but as fibres containing AXs (such as bran). Therefore, AX’s benefit for human health deserves further investigation. The relationship between arabinoxylan structure and its physicochemical influence on cereal products (pasta, cookies, cakes, bread, and beer) is also discussed. A strong correlation between arabinoxylan’s structural properties (degree of branching, solubility, and molecular mass) and its functionalities in food systems can be observed. There is a need for further studies that address the health implications behind the consumption of arabinoxylan-rich products. Indeed, the food matrix may influence the effects of arabinoxylans in the gastrointestinal tract and determine which specific arabinoxylans can be included in cereal and non-cereal-based food products without being detrimental for product quality.

## 1. Introduction

In recent years, the link between diet and human health has become more widespread. The positive effects of fibres on the population’s health are becoming more apparent, although there is still more to learn. A clear relationship between the different characteristics of fibres (e.g., structure, solubility, viscosity, etc.) and their health benefits is still unclear [1]. Nevertheless, there is a consensus that fibres act as prebiotics and positively affect human health. Prebiotics are compounds that stimulate bacteria’s growth and correlate with reducing diseases caused by a shift in the microbiota [2,3]. This effect has been proven in different studies, such as the ones conducted by Paesani et al. [4], Nguyen et al. [5], and Carvajal-Millan et al. [6], among others. Many fibres have a prebiotic effect [1]. Among these are arabinoxylans (AXs). AXs are polysaccharides present in the cell walls of different plant tissues. They are composed of a linear backbone of xylose units linked by β1-4 bonds with arabinose units linked to some of the xylose units. Moreover, the xylose units can carry methyl-glucuronic acids, and arabinose units can bond to ferulate residues [7]. However, the structural characteristics of different AXs are complex and influenced by their source. AXs present a wide range of water solubility (soluble and insoluble AXs), which depends on factors such as the average degree of polymerisation (DPav), degree of branching, and monomeric composition [8,9]. Their ability to link to other polymers found in plant tissues (such as hemicellulose) also influences their solubility, reducing their extractability with water [10]. 

This review aims to gather the most critical and relevant research surrounding AXs and go through their structure, health benefits, and current cereal-based food product application to better understand their complexity and the benefits that can arise from their use in food products for human consumption.

## 2. Structure of Arabinoxylan

As previously stated, arabinoxylan (AX) is a polysaccharide present in the cell wall of various cereals, such as wheat, corn, rye, barley, rice, and oat [11], and it is composed of a linear backbone of xylose units with linked arabinose units. More precisely, the general structure of an AX consists of a linear β-(1→4) linked xylan backbone to which α-l-arabinofuranose units are attached as side residues via α-(1→3) and/or α-(1→2) linkages [12]. The molecular structure of AX is also dependent on the extraction method applied. AXs can be extracted using chemical, enzymatic, or physical treatments [13]. A wide range of AXs are found in different plants; this review will discuss the structure of AXs present in three primary sources: wheat, barley, and corn. Table 1 summarises the structural characteristics of AXs from these main cereal grains, including wheat, barley, corn, rice, sorghum, rye, and oat.

### 2.1. Structure of Wheat Arabinoxylan

Wheat AXs are present in endosperm (3–5% of total endosperm), aleurone, and bran cell walls (approximately 60–70% of the entire cell wall) [12,15]. In the specific case of wheat bran, AXs represent between 10.9 and 26% of all the bran fractions [37,38,39,40].

In wheat AXs, side chains are linked by α-(1→2) and/or α-(1→3) bonds along the xylan backbone. The xyloses can be di-substituted, mono-substituted (the most common substitution), or not substituted at all [16,17]. These side chains are mainly formed by single arabinose units (α-l-arabinofuranose), but side chains linked to xyloses of α-d-glucuronic acid (and its methyl ether, 4-O-methyl-glucuronic acid) also occur [16]. The structure of wheat AXs presents a wide variability, as reported by several authors [40,41,42]. These differences are influenced by the wheat variety and the wheat grains’ maturation stage. It has been reported that the arabinose/xylose ratio decreases upon maturation [43,44], having a positive influence on wheat AX’s water solubility [45] According to Barron et al. [39], AXs of the endosperm present a higher water solubility than AXs from bran, as well as a lower arabinose/xylose ratio (A/X) (~0.6) than that of AXs derived from bran (~1). Other studies confirm these findings [46,47,48]. However, Kaur et al. [40] reported A/X ratios to be considerably lower than 1 for wheat brans of four different wheat varieties (between 0.33 and 0.62). These authors also reported different A/X ratios for bran fractions rich in AXs. They found A/X ratios between 0.09 and 1.37 (for water-extractable fractions), 0.33 and 1.82 (for alkali-extractable fractions), and 0.38 and 0.7 (for cellulosic arabinoxylans). This variability is a good indication of the complexity and variability of wheat AX’s structure. However, it seems clear that the A/X ratio is lower for AXs located in the endosperm than for those located in other parts of wheat grains. The A/X ratio plays an important role in modulating the hydration and swelling capacity of AX [49]. Maes and Delcour [50] observed that wheat AX extracted from wheat bran had an A/X ratio of 0.45, but the gradual precipitation of AX with ethanol changed the ratio significantly from 0.31 to 0.85, depending on the percentage of ethanol used, demonstrating the influence that the type of extraction method can have on the A/X ratio. In addition to the xyloses, arabinoses and α-d-glucuronic acid units that form part of the AX’s other short sugar side chains can also be present in wheat AX’s structure. These side chains are constituted by xylopyranosyl and galactopyranosyl residues associated with arabinofuranosyl residues [16]. Additionally, arabinose units/chains can also carry acetic acid and hydroxycinnamic acids (ferulic and p-coumaric esters) [16,45].

### 2.2. Structure of Barley Arabinoxylan

The basic structure of barley AXs is the same as that of wheat AXs (polysaccharides mainly composed of xylose and arabinose). However, there are some notable differences. For example, barley AXs present side chains of xylose units in the 2 and/or 3 C of the xyloses, forming the backbones of AXs [19,20]. On average, barley AXs have a higher A/X ratio than wheat AXs [51], since their arabinose side chains are more numerous. The molecular weight (Mw) of barley AXs is also distributed in a wide range for kDa [19,22,52], having a higher Mw for water-soluble AXs [22]. Barley AXs are distributed along all the grain, representing around ~10–14% and ~1.2–1.3% of the bran fraction and endosperm, respectively [20,22], and around 25–40% of barley cell walls [8]. Evidence supports a positive relationship between higher A/X ratios (implying more branching) and improved water solubility. Izydorczyk et al. [20] reported both AX’s higher solubility and higher A/X ratios (from the water-soluble AXs) from bran fractions (~0.8–1) than from an endosperm fraction (~0.65–75) [20]. However, when comparing the A/X ratio of water-soluble and -insoluble AXs, these authors observed that insoluble AXs from the endosperm had a higher A/X ratio than that of soluble AXs. In disagreement with these results, Lazaridou et al. [8] reported a higher A/X ratio for water-soluble AXs than for non-water-soluble AXs originating from the endosperm. These differences between studies could be related to the barley variety investigated, the DPav of the AXs, the germination state, or the nature of the other polymers in the grain, among other causes. In such regards, Izydorczyk et al. [20] found a relationship between starch structure and AX’s solubility, reporting a positive relationship between the water solubility of these carbohydrates and the amylose content of the starch of barley grains. In addition, these same authors reported AXs with higher ferulic acid content in high amylopectic grains.

### 2.3. Structure of Corn Arabinoxylan

Corn is also a good source of AXs, although it is much less studied than AXs from wheat or barley [24,53,54]. Around 51% of corn bran has been identified as AXs, or 67% if residual starch is not considered [24]. However, other authors have reported lower yields of AXs from corn bran (around 35–40%) [55,56]. These AXs have a highly branched structure with a xylose backbone and arabinose residues as side chains on primary and secondary hydroxyl group structures, with an A/X ratio of around 0.6 [24]. Glucuronic acid (linked to the o-2 position of the xylose forming the backbone), galactose (linked to the arabinose branches), and some xylose residues also form part of corn AX’s structures [24,25,26]. In addition to this, p-coumaric acid, ferulic acid, and acetic acid have also been found to be esterified to the monomers forming the corn AXs [24].

## 3. Extraction and Production of AXs as a Food Ingredient

Extraction of AXs from cereals can be performed using various techniques from different parts of the grains. The most common source from which AXs are extracted is cereal brans, where the concentration of AXs is greatest (between 10 and 25% of the total bran) [20,22,37,39,40]. Extraction of AXs can be performed by water treatments, mechanical treatments, chemical treatments, enzymatic treatments, or by combining these techniques [12,13,42,57,58]. Figure 1 illustrates the different treatments that can be performed for AX’s extraction, including water and chemical treatments and other mechanical approaches.

### 3.1. Water Extraction of Arabinoxylans

Water extraction of AXs is the easiest and least aggressive extraction method capable of preserving AX’s native structure. As previously discussed, the water solubility of AX is dependent on several factors, such as the type of grain, the degree of germination, and the nature of the polymers forming the grain [8,9]. These factors will undoubtedly impact the yield of AXs when extracting with water.

The extraction procedure involves solubilising the AXs by placing the milled grains (or grain fractions) in water at temperatures that can range from 45 to 90 °C for a fixed time (usually longer when using lower temperatures) [8,43,59,60]. This solution will then be precipitated using an organic solvent. To inactivate the grains’ endogenous enzymes, samples can also be pre-treated with an aqueous ethanol solution (80% *v*/*v*) [8]. After extraction, insoluble polymers are removed by centrifugation. The supernatant rich in AXs can be directly lyophilized to retain a pellet rich in AXs [8] and other water-soluble polymers. To overcome this, an alternative step following the first centrifugation can be performed. The AXs in the supernatant can be precipitated with 95% ethanol or another organic solvent at around 4 °C for a fixed time (typically 12 h), followed by centrifugation and drying steps [4,60]. Before measurement, the lyophilized sample can be treated to remove denatured proteins by filtration with celite or an equivalent compound (e.g., Fuller’s earth), and by adsorption on Vega clay (or equivalent) for the residual non-denatured proteins [50,59,61]. Depending on the raw material used for the extraction, removing other polymers such as starch and other carbohydrates may be required. Removal is typically achieved using specific enzymes that target these polymers. Free sugar is then removed using dialysis while the enzymes are heat-inactivated [59]. These proteins and non-AX carbohydrate removal steps can also be achieved before the lyophilisation of the pellet rich in AXs [59]. The main limiting factor of these extraction methods is that the crosslinks between potentially soluble AXs and other polymers of the cell wall matrix are not broken, limiting the extraction yield [12]. Thus, it might be more appealing to couple water extraction of AXs with mechanical treatments to increase solubility. The following paragraph reviews the most critical mechanical treatments to improve AX’s extractability. 

### 3.2. Mechanical Extraction of Arabinoxylans 

Mechanical extraction helps to improve the extraction yield by making AXs more accessible. In addition, other mechanical treatments are available, such as milling and extrusion [62,63,64,65], ultrasound [66,67,68], microwave [69,70,71], or steam-pressure [64]. Milling and extrusion of cereal flours/bran before an AX extraction can increase the yield of water-extractable AXs. However, such mechanical treatments can also affect the structure of AXs, reducing the substitution degree of AX significantly [64]. The application of ultrasound technology is another successful approach for AX extraction. This technology can substantially reduce the time required (from hours to minutes) to achieve a targeted yield [66,68]. It is essential to control the power used, as relatively high ultrasonic power can negatively affect the extraction yield [68]. Microwave technology is a unique approach that can improve AX extractions [69,70,71]. Compared to conventional heating methods, microwave-assisted extractions can help reduce extraction times, increase efficiency, reduce solvent consumption (if applied), and lower energy requirements [72]. Davis et al. [72] recently reported that the microwave extraction of polysaccharides could affect the Mw distribution of the extracted carbohydrates and the relative abundance of different polysaccharides in the final extract. Unfortunately, there is not enough research on the effect of microwave extraction of AX, as available literature often focuses on polysaccharides as a whole. Hence, more research is required to better understand the impact of microwave extraction on AX. Steam-pressure application to stabilize against spoiling flours or bran can also positively affect AX’s extractability. For example, according to Kong et al. [73], steaming bran improved soluble fibre extractability as the soluble dietary fibre percentage increased from 4.57 to 9.10%; this was in agreement with Aktas-Akyyildiz et al. [74], who reported an increased water extractability of AX (from 0.75 to 2.06%) after steam-pressure treatment. Similarly, Sui et al. [75] reported that steam pressure could transform some insoluble dietary fibre into soluble fibre, thus improving water extractability. Another approach that is becoming more common in the literature is the combination of different mechanical treatments [76,77,78]. However, there is still a lot to be understood about the combination of different mechanical treatments for better AX extractability, as research related to this is still limited. 

### 3.3. Chemical Extraction of Arabinoxylans 

Chemical extraction of arabinoxylans can be performed using alkali or acidic solution and has been well-reviewed by Zhang et al. [42]. The chemical extraction procedure of AX consists of submerging the raw material in the chemical solution and extracting for a set period using specific conditions. After extraction, the solid residue needs to be separated, which is achieved by centrifugation [57,79]. When the extraction is performed using an alkali solvent (sodium hydroxide (NaOH)), the pellet is washed several times to remove undesirable compounds. The solids are then dried to obtain the AX’s rich fraction [57]. When using an acid solvent, the pellet is discarded, and the supernatant is treated with three times the volume of 95% ethanol to achieve hemicellulose precipitation. The pellet is then separated by centrifugation and washed with acidified 70% ethanol before drying to obtain an AX’s rich fraction [79]. Depending on the chemical used for the extraction, these steps might vary. Alkali solvents can disrupt covalent and hydrogen bonds and loosen up cell wall matrixes, which results in a release of polysaccharides present in the cell wall that cannot be extracted with just water [80,81]. Alkaline solvents can also change the charge of uronic acid residues to their negative form, favouring repulsion forces that improve AX extractability [82]. Table 2 summarises some of the available studies that use different alkali solvents to extract AXs. Acid solvents for AX extraction are not as common as alkali solvents because they can have a hydrolysing effect on the AXs of interest. If the chemical hydrolysis is extensive, some AXs may be degraded into very low-Mw AXs that dissolve in the organic solvent, leading to a decreased yield [42,81]. 

### 3.4. Enzymatic Extraction of Arabinoxylans 

Enzymatic extraction of AXs with the use of endoxylanases and cellulases can be as efficient as chemical methods, with the benefit that it is more environmentally friendly and AX degradation can be better controlled. Treatment conditions influence the yield, Mw, and A/X ratio of extracted AXs. The enzymatic effect on the AX extraction yield is influenced by the enzyme source and concentration, and it depends on whether they are used alone or in combination with another enzyme (Table 2). The combined use of endoxylanases and cellulases provides higher extraction yields of AXs [91,92]. A more common approach is to couple an enzymatic extraction of AXs with other extraction methods, typically chemical extractions with alkali solvents and chemical treatments [13,66,69]. The enzymatic extraction can be performed after extracting the water-soluble AXs of the raw material to maximise the yield of AXs. 

## 4. Health Benefits

Research on this topic has alluded to the health benefits of following a fibre-rich diet. Reynolds et al. [93] performed a systematic review and meta-analyses of the available studies that address relationships between carbohydrate quality and non-communicable disease incidence, mortality, and risk factors. According to the available data, increasing daily dietary fibre intake reduces the risk of cardiovascular-related issues, diabetes, and cancer. In addition to this, emerging evidence has shown that dietary fibre consumption patterns are linked to improved mental health [94,95] and other cognitive functions [96]. Because of this, it is vital that a better understanding of the mechanism behind fibre’s specific health benefits is established. In the particular case of AX, a prebiotic effect has been reported in in vitro and in vivo studies. In a recent study by Lynch et al., [97], brewer’s spent grain (BSG) was used to extract soluble AX through simultaneous saccharification and fermentation processes. The extracted solutions containing 99% soluble AX were studied for their microbiome-altering abilities through in vitro faecal fermentation trials. The authors found the extracted AX showed prebiotic effects resulting from the 2-fold and 3.5-fold increase in *Lactobacillus* and bifidogenic levels, respectively. In such regard, He et al. [11] postulated that the regulation of intestinal microflora is positively correlated with the degree of aggregation and branching of the AX’s propionate and acetate, (which have been associated with positive health outcomes [1]) that were also detected in the samples containing the highest levels of soluble AX. As previously mentioned, the chemical structure of AX is not straightforward and is influenced by the source, germination state, location within the grains/plant material, and the extraction process. Since the chemical structure of fibres can affect their physicochemical characteristics and thus determine their functionality in the gastrointestinal tract [1], it is impossible to hypothesise the exact effect of all AXs. Therefore, to better understand their impact on the gastrointestinal tract, the chemical structure of the specific AXs being studied must be understood. 

Another issue regarding the explored prebiotic effect of AXs is that the available studies are either in vitro or in vivo, with only a few studies that apply both approaches [4]. Since in vitro studies investigating the effects of fibre do not always correlate with the observations in in vivo studies [1], it makes it even more challenging to reach a consensus regarding their effect. Another variable is that in vivo human trials typically rely on the inclusion of AXs in the diet through supplements, which omits considering how the food matrix can impact the effect of AXs. It is also important to note that AX structure can affect the physicochemical characteristics of different food products, as discussed later in this review. Thus, it can be the case that AXs with promising health benefits (as reported in clinical studies) cannot be easily incorporated in the food matrixes without significantly reducing the quality and characteristics of the food, influencing, thus, the consumers’ perception. In addition to this, the available clinical studies involving humans do not control the participants’ diet entirely. To the best of our knowledge, there are no such studies with AX fibres. Walker et al. [98] evaluated the effect of corn bran that contains AXs in a human intervention trial with fully controlled dietary intake. However, since AXs were not tested alone, their observations cannot be merely associated with the presence of AXs in the food matrices.

An important variable to consider with the in vivo studies is the initial microbiota of the participants. A microbial community that lacks essential bacterial species involved in the breakdown of complex polysaccharides is likely to be less efficient at degrading these polymers [99], influencing the studies’ outputs. This was proven by Wu et al. [100], who found entirely different results for groups with different geographical locations. However, despite all this, there is a clear consensus regarding the main beneficial effects of AXs [13]. These are as follows (Figure 2):modification of short-chain fatty acids (SCFAs) production in the colon via regulation of gut microbiotaantioxidant capacityhypoglycaemic effect/postprandial blood glucose response control

Table 3 shows some of the most recent and relevant findings regarding the effect of AX chemical structures on regulating these effects. SCFAs ratio and abundance correlated with the gut microbiota composition, which is influenced by the chemical structure of the AXs used in the studies [5,13,101,102,103]. In in vivo studies, data on the concentration of SCFAs should be considered with great vigilance, as concentrations of faecal SCFAs are affected by the absorption capacity of the epithelium and the AX consumption [104]. Similar observations were reported by Bach Knudsen et al. [5] in an in vivo study with pigs. This could result in a lack of significant differences in total SCFAs concentrations in the group consuming AXs and the control group, as reported in the study of Nguyen et al. [5]. However, other in vivo and in vitro studies have reported an increase in the total concentration of SCFAs (Table 3) [101,102,105]. In a human intervention study, Walton et al. [105] studied the effects of consuming AX-oligosaccharides enriched bread. They found the faecal butyrate content and the levels of faecal bifidobacteria to increase. The antioxidant capacity of AXs seems to be influenced by the state (bonded or free) [11] in which ferulic acid is present within the AXs, with higher antioxidant capacity observed when bonded [13,106]. Mw and the degree of substitution of the AXs also influence the antioxidant capacity. AXs with lower Mw and degree of substitution seem to have improved antioxidant capacity [13,101]. On the other hand, these same authors found that the hypoglycaemic effect of AX is better when the Mw and the degree of substitution are higher [101]. These findings by Chen et al. [101] are fundamental, as they reveal that a particular AX can be more suited for one specific health benefit than another.

The hypoglycaemic effect of AX relates to the modulation of the α-amylase activity, which can break down long-chain carbohydrates with a subsequent increase of the postprandial blood sugar levels [13]. Chen et al. [13] found a link between the AX extraction method and its hypoglycaemic effect by reporting a higher α-amylase inhibition for alkali-extracted AXs (containing a higher free ferulic acid amount). Other studies report that free ferulic acid has a higher inhibition power than esterified ferulic acid in rats’ intestines [108]. Hartvigsen et al. [109] and Boll et al. [110] also reported that AXs positively affected blood glucose response in rats and humans, respectively, but these authors did not refer to ferulic acid. AXs can also influence the postprandial blood glucose response through their viscous properties. Vogel et al. [111] reported decreased blood glucose levels in rats fed with cross-linked AXs, although no effect was observed when fed with native AXs. Due to the above-mentioned health benefits of AXs, there is an increasing interest in their incorporation in different foods and beverages. However, the number of studies focusing on adding AXs in various food matrixes is still scarce. Moreover, most of these studies do not investigate the effect of AXs as a pure ingredient on food systems but as fibres containing AXs (such as bran). 

## 5. Physicochemical Properties of AXs

### 5.1. Solubility 

As previously stated, AX solubility is affected by the specific structure of the AX, the extraction procedure/treatments used, and the type of linkage to other plant cell tissues [8,16,20,112]. According to the existing evidence, water-soluble AXs are usually highly branched [112,113], probably because solvated AX flexibility is better when highly substituted [114]. In the late seventies, Andrewartha et al. [115] discovered that the water solubility of AXs treated with arabinofuranosidase resulted in AXs with an A/X ratio of 0.43 and a water solubility of ~70%, wheras the unmodified AXs had an A/X ratio of 0.50 and were completely water-soluble. When low-substituted AXs occur (low A/X ratio), they tend to self-aggregate to have more favourable configurational entropy, explaining the decrease in the AX solubility [114]. However, branching is not the only factor that affects AX solubility. Dervilly et al. [116] reported that AXs with different Mw but similar substitution levels (around 40% of xylose units substituted, mainly disubstituted) showed dissimilar behaviour. AXs with lower Mw tended to aggregate, which indicates that a decrease in solubility had occurred. According to Pitkänen et al. [117], another aspect to consider for AX solubility is how arabinose residues are linked to the xylose chain. These authors found that when the A/X ratio is reduced through the action of enzymes and the removed arabinose units are from disubstituted xyloses, the solubility of AX remains high. On baking application, AX solubility seems to exert a critical role mainly due to the insoluble AX aggregates that cause an uneven dough mixing, compromising the stability of the bubble interface [111].

### 5.2. Viscosity 

The viscosity of AX is positively correlated with solubility because of the high water-holding capacity of soluble AX [118,119]. Like solubility, Mw also plays an essential role in influencing the AXs’ viscosity [112,116]. AXs with similar branching [112] or similar xylose substitution levels [116] and lower Mw resulted in lower viscosity profiles. Kale et al. [120] and Đorđević et al. [121] also reported lower viscosity levels for AXs with lower Mw after treatment with enzymes. For AXs with a similar Mw, branching is also an important characteristic to consider. According to Kale et al. [122] AX branching also affects the viscosity behaviour of AXs. These authors found that, for extensional viscosity analysis of AX, highly branched AXs showed a less pronounced viscosity decrease when increasing the flow rate. This is probably because polymers with higher branching levels can lead to greater entanglement, resulting in higher resistance to extension [123]. These findings were also confirmed by Pavlovich-Abril et al. [112] by suggesting that high intrinsic viscosity correlates with low A/X ratio and high elongational viscosity of dough.

### 5.3. Emulsifying Capacity 

Some studies reported that AXs have emulsifying capacity; however, much of the emulsifying ability of AX has been attributed to phenolic groups or protein links to AXs [124]. Yadav et al. [53] compared laboratory-extracted and industrially produced corn AXs and acacia gums (known to have emulsification properties) for their emulsification properties. The authors observed that emulsions prepared with some AXs had better emulsion stability over time than emulsions prepared with the acacia gums. Proteins were present in all the purified AX samples, and those with higher protein content had better emulsification properties. In a subsequent study, Kokubun et al. [125] showed that high-Mw AXs with a significant amount of proteinaceous material absorbed have better emulsification properties, suggesting a synergic effect between AXs and proteins. Xiang & Runge [126] evaluated the emulsifying properties of arabinoxylan-protein gum (APG) (before and after a succinylation process) and compared it with arabic gum. These authors showed that emulsion particle size and stability of AX-protein gum and gum arabic were comparable at pH 3.5–6.5. The succinylation process enhanced the emulsifying properties of APG. Compared to gum arabic, at pH < 5, the succinylated APG (SAPG) emulsions had larger particle size but comparable stability, whereas at pH > 5, SAPG had much smaller particle size and better stability than gum arabic. However, emulsions stability was influenced by the pH and the succinylation process. Kale et al. [120] evaluated the emulsification properties of AXs after enzymatic treatments. These authors observed that AX’s Mw did not influence the emulsification capacity of AX. However, emulsion stability was an influential factor, as the mean droplet size after 3 and 7 days of storage at 60 °C was higher for the hydrolysed AXs [120]. This increase in mean droplet size can result from coalescence or bridging phenomena; however, the authors provided no additional information on this topic. Another aspect to consider when evaluating a particular molecule’s emulsifying properties is the resultant emulsion’s final viscosity, as this will determine its potential application. According to the results available, applying AXs with lower Mw results in emulsions with lower viscosity [120] Succinylation of AX-protein gum also decreased the viscosity of emulsions [91].

## 6. AXs Inclusion in Food Matrixes

Studies that focus on the incorporation of AXs in food matrixes are scarce. However, some research articles focus on modifying intrinsic AXs. The information from these articles can elucidate what to expect when incorporating AXs and the potential challenges associated with this. The most relevant information on AX functionalities in different food and beverage products are summarised hereafter and in Table 4.

### 6.1. Pasta

A few studies explore the functional properties of AXs in pasta. Ingelbrecht et al. [128] investigated the influence of arabinoxylans and endoxylanases on pasta processing and quality, aiming to produce pasta with increased levels of soluble fiber. They observed an increase of solubilised AX (as an additional source of soluble dietary fiber) of low molecular weight and low leaching out (maximally 5.9%) during the cooking process of pasta. In another investigation, Ingelbrecht et al. [128] quantified the solubilisation of AX during different pasta processing, characterising the cooking losses after optimal and excessive cooking times of the pasta samples. They observed significant solubilisation of AX during pasta processing and minimal changes in the WE-AX’s structural properties (Mw profile and substitution pattern). They related the changes in AX’s structure to the mechanical forces (e.g., extrusion/lamination), as endogenous endoxylanases activity was observed to be minimal. Again, at optimal cooking times, very low losses of WE-AX (1.48–3.64%) were noticed as a source of soluble dietary fiber, and excessive cooking times resulted in significantly higher AX losses (5.02–48.47%).

In a different study, the same research group studied the effect of exogenous enzymes (endoxylanases) on pasta AXs [129]. They observed that the extrusion pressure was reduced due to the enzyme addition. AX loss at the optimal cooking time increased proportionally to enzyme dose, but is less than expected (control pasta had an AX loss of 2.2%, whereas enzyme addition at different levels resulted in a loss of 2.1–19.8%). Overcooking resulted in AX loss of up to 45%. The quality of the pasta was also affected by these treatments (pasta was softer and more fragile), which suggests that when applying enzymes, lower hydration levels should be used to achieve a high-quality pasta [162].

AX fortification strategies have also been investigated in different cereal-based products. For example, Turner et al. [130] studied the effect of water-soluble AXs to fortify pasta. The fortification with up to 2% *w*/*w* of WEAXs resulted in a linear increase in farinograph water absorption (around 5% increase for every 1% increase of AXs) [130], leading to dough weakening. In addition, reduced cooked pasta stickiness was observed at all levels of WEAX addition. In a later study, Ciccoritti et al. [168] studied the effect of semolina substitution by different bran fractions (coarse and fine fractions) to naturally increase the potential nutritional value of pasta. The pasta enrichment was attained by adding two different ratios of coarse and fine flour fractions (25 and 50 g/100 g, respectively). They reported an increase in the AX content of up to 64% in arabinoxylans, reaching values of 3 g/100 g, along with good cooking quality.

### 6.2. Cookies and Biscuits

The effect of AXs in different cookie samples has been reported in the literature. Regardless of the cookie type, the use of flours with higher AX levels results in cookies with a lower spread ratio, decreased diameter, and harder texture [131,132,134,135,137], which is detrimental for their quality. However, Guttieri et al. [132] also reported that higher A/X ratios were correlated with a greater spread than AXs with lower A/X ratios, as highly branched AXs tend to reduce water absorption in whole grain flour, resulting in more tender cookies [137]. Pareyt et al. [136] studied the effect of flour or sugar substitution in sugar snap cookies by AX oligosaccharides with a low A/X ratio (0.2). When replacing high percentages of flour with AX oligosaccharides, the authors observed that the dough’s water level had to be reduced to obtain a workable dough (without water reduction, the dough was too sticky). Most probably due to their low Mw, AX oligosaccharides increased spread ratio but resulted in cookies with an unacceptable structure [136]. When using these AX oligosaccharides as sugar replacers, Pareyt et al. [136] reported that cookie diameter was slightly decreased, and hardness was moderately increased. This seems like an exciting strategy to increase fibre content while reducing sugar levels. Heredia-Sandoval et al. [133] studied the effect of a brewer’s spent grains on cookies and reported increased AX levels, decreased spread ratio, and increased hardness for enriched cookies. These findings were in agreement with previous studies. 

### 6.3. Cakes

The effect of AXs in cakes has not been widely studied. Some studies have referred to the potential benefits of AXs, such as the one carried out by Oliete et al. [169] that suggested that flours with good baking performance for cake elaboration can be selected based on higher soluble pentosan concentration. However, to the best of our knowledge, no specific research on the effect of AXs on cakes is available. Lebesi and Tzia [104] reported improved quality of bran-enriched cakes when using xylanases. This suggests that the chemical structure of AX modulates product outcome, as observed in other types of matrixes. However, no information on AXs was reported in the article. In a later study, Moza and Gujral [166] reported that flours with a high content of AXs promoted smaller air cells in final cakes. Specifically, they concluded that the presence of β-glucans and arabinoxylans exerted multi-beneficial effects on batter consistency and cake cell density and crumb uniformity, improving the water binding capacity simultaneously, thus keeping the cake crumb moist. In a recent review, it was reported that during cake-batter mixing, the sugar water syrup acts as a solvent to increase the beneficial functions of the endosperm arabinoxylans, gliadins, and expectedly PINs, while acting as a plasticizer to decrease the detrimental function of the glutenins. Later, Haghighi-Manesh and Azizi [170] evaluated the effect of extrusion and enzyme treatment of bran and its later incorporation in cakes and observed that adding modified bran in cakes resulted in products with better sensory attributes. The beneficial effect of extrusion on whole-grain flour on sensory and total and soluble dietary fibre levels has also been reported for whole corn flour by Paesani et al. [171]. Therefore, the effect of AXs on cake characteristics should be investigated further.

### 6.4. Bread

The use of arabinoxylans in bread can positively influence the final product characteristics, as previously reviewed by Saeed et al. [172]. However, since then, significant contributions have been made by the research community, as recently reported by Pietiäinen et al. [173]. Small additions of AXs have been shown to positively affect the physicochemical properties of bread, such as increased specific volume or controlled starch retrogradation [162,172,174,175,176]. However, the addition of high doses of AXs can have a detrimental effect (e.g., low specific volume or poor texture) [162,176]. The chemical structure of AXs and any potential structural changes caused by the bread-making process also need to be considered when evaluating the effect of AXs in bread [177]. For example, according to Labat et al. [178], the percentage of intrinsic soluble AXs increased during dough mixing, but the increase was higher when ferulic acid was added to the dough. The ferulic acid content of AXs also seems to influence dough development by increasing dough extensibility without displaying differences in the volume of the bread (compared with AXs with lower ferulic acid levels) [160]. The most common approach to overcoming the drawbacks of the effects of large doses of AXs is the addition of enzymes targeting them [179]. This is done because Mw and A/X of AXs are, as in other products, important parameters to take into consideration when evaluating AX’s effect on bread. AXs with lower Mw seem to show better results in final bread [161], although their Mw also influences the characteristics (and therefore the handling) of the bread doughs [180,181]. The use of enzymes, both for intrinsic or added AXs, can improve bread quality (e.g., increased volume and retarded bread staling) while also improving the total soluble AXs in the final bread [140,142,144,147,149,153,154,160,161,182,183,184]. However, it should be taken into consideration that the use of higher enzyme concentrations that target AXs can result in an increased dough stickiness because of the decrease in AX’s Mw. Courtin et al. [143] dealt with this problem by adjusting the water level by hand, as performing it via farinograph resulted in worse consistencies. Differences are also observed within studies that test different enzymes [143,144]. Xue et al. [160] revealed how minor chemical differences of AXs can affect bread doughs and the final bread. Undesirable effects must also be considered when applying enzymes to improve AX’s performance in bread. Evidence correlates the presence of xylanases with syrup formation in refrigerated doughs [145,156]. This undesirable syrup formation may be linked to a decrease in the Mw of AXs upon storage time [155] and the effect of starch and proteins interacting in the dough system. Another aspect to consider when evaluating the effect of AXs on bread properties is the addition of other unrelated enzymes (not targeting AXs). The use of glucose oxidase has been shown to catalyse the formation of AX–AX crosslinks and crosslinking between AXs and other polymers. Crosslinked AXs have been shown to have negative effects on bread characteristics in various studies [185,186]. However, glucose oxidase has also been shown to positively impact other polymers [187,188]; therefore, a balance must be met to achieve maximum quality. In disagreement with these authors, C. Zhang et al. [162] have reported that the addition of lipoxygenasenhanced crosslinking between water-soluble AXs positively affected bread quality. These different results could be related to the specific structure of the AXs of each study. Another enzyme that probably favors the cross-linking of water-soluble AXs is laccase [189]. Though the effect of this enzyme may not be determined for intrinsic AXs in the flour, their use should be taken into consideration when incorporating AXs. AX structures can alter their effect; therefore, incorporating these enzymes can lower A/X ratios and induce crosslinking with other polymers present in bread (such as gluten or starch) [180]. In addition to utilising these enzymes to overcome the drawbacks of AXs, fermentation [138,142,150,152], a combination of enzymes, extrusion methods [157], and sprouting [141] have also been proposed for added rich bran fractions. Fermentation and sprouting seemed to improve the performance of bran-enriched bread, resulting in increased soluble AXs, good textural properties, and improved volume. Liu et al. [112] studied the effect of ultrasonication, enzyme (xylanase), and trifluoroacetic acid treatments to modify AXs and examine their effect on dough characteristics to later include them in bread by replacing 6% of the flour. These authors concluded that all treatments reduced the Mw of AXs and that ultrasonication plus enzyme treatment was the best combination to improve the GI response of bread consumption. However, the effects of this treatment on final product characteristics were not reported.

Apart from the health benefits of AXs inclusion on bread, AXs may also positively affect frozen doughs. Adams et al. [172] reported that bread doughs with increased AX content could be baked after being frozen without a significant loss on the specific volume of the final bread, whereas control bread suffered a considerable volume reduction when baked with frozen doughs. These results are backed up by P. Wang et al. [159], who evaluated the effect of AXs on the quality of frozen steamed bread dough. This positive effect of AXs on frozen bread doughs could be related to changes in water-soluble AXs during storage at freezing temperatures, increasing the viscosity of the aqueous phase of dough, and providing increased stability to the system during frozen storage [139]. 

The presence of AXs in bread also influences other polymers (such as starch or gluten) present in the doughs. AXs have been shown to decrease the glass transition temperature of wheat doughs for a wide range of water activities [190]. According to the findings of Qiu et al. [183], AXs can modulate starch, leaching to a result of starch granules being more compact in the presence of AXs (as shown using confocal laser scanning microscopy (CLSM)). The same authors observed that the presence of AXs also lowered peak viscosity and breakdown during starch gelatinization. Final viscosity was found to increase, likely due to the interactions between AXs and amylose. P. Wang et al. [191] and Whitney and Simsek [192] also reported a reduced starch gelatinisation in the presence of AXs, as well as a reduced glycaemic index [192]. Wang et al. [191] and Hou et al. [148] concluded that AXs with lower Mw hindered starch gelatinisation more evidently. The same authors also concluded that the Mw of AXs influenced how they suppress gelatinisation: low and high-Mw AXs retarded amylose and amylopectin recrystallization, respectively, revealing high-Mw AXs to have a more significant contribution for long-term starch retrogradation. 

The addition of AXs in baking formulations have been investigated for their ability to interact with various macromolecules such as proteins and other carbohydrates. A study by Buksa et al. [185] explored different approaches to improve rye bread using various concentrations of AX (cross-linked and hydrolysed), starch, and proteins isolated from rye wholemeal as substitutions for rye flour in a model system. The authors established a 6% hydrolysed AX and 3 and 6% protein formulation to produce an optimal product with improved water absorption, dough yield, bread volume, and reduced crumb hardness [185]. Döring et al. [193] also showed that 2.5% AX concentration in a rye model dough resulted in a greater distribution of protein formation, leading to a reduction in dough elasticity and a positive effect on the volume and crumb hardness of the final bread. However, the authors also observed that an increase in AX to >5% prevented protein formation and resulted in a detrimental effect on the end product. AXs have been shown to interact with other AXs and starch, altering the rheology of wheat/rye bread dough [194]. A study by Buksa and Krystyjan [195] found that the baking process of rye dough composed of hydrolysed AX, protein, and starch resulted in a limited amount of starch swelling, imparting a high-quality crumb formation, compared to cross-linked AX complexes that revealed the opposite effect. 

Throughout the years, numerous studies have tried to understand the mechanism of action of AXs on gluten proteins. M. Wang et al. [158] reported that water-soluble pentosans (which encompass AXs) interfere in developing the gluten network, resulting in the formation of a less extensible network. However, according to the same authors, this can be prevented by adding ferulic acid or xylanases. Santos et al. [151] and Q. Li et al. [151] revealed a reinforcement of the gluten network with pentosans/AXs, but no information about the presence of ferulic acid was reported. The positive effect of ferulic acid was observed by P. Wang et al. [191]. Q. Li et al. [186] found differences between the soluble fibre structure (mainly formed by AXs) and the effect on the microstructure on the dough. They noted that high Mw fibres with a low level of ramification resulted in fibre aggregation insertions in the dough network, and fibres with lower Mw and higher branching (high A/X ratio) diffused better in the dough network. Q. Li et al. [151] hypothesised that the fibres with lower Mw and higher branching would give rise to a relatively stable dough structure with stronger gas retention capacity and therefore better dough performance. This study agrees with the observations of the previous research that found better baking performance when using AXs with lower Mw. In a later study, Zhu et al. [196] found that a negative effect on elasticity was driven by high-Mw water-soluble AXs, and that those with lower Mw integrated better in the matrix and even improved the gluten network functionality, as also reported by Döring et al. [146] and Zhao et al. [148]. These results again show the importance of fine structure when evaluating the effects of AXs in cereal-based products. 

Gluten proteins are also affected by temperature increases, as the temperature has a polymerizing effect that induces covalent bonds between proteins [197]. According to Santos et al. [198], gluten is less sensitive to heat in the presence of pentosans. Noort et al. [199] showed that fibre’s general negative effect on bread quality is more related to fibre–protein interactions than gluten content dilution. Zhu et al. [200] confirmed that water-soluble AXs prevent chemical changes in gluten during heating and reduce gluten elasticity at different temperatures. Nevertheless, and as stated before, the effect is dependent on the AX’s Mw [146,196]. Zhao et al. [162] reported that water-soluble AXs induced a higher aggregation trend of gluten upon heating (enhancement of the glutenin-gliadin aggregation at 90–95 °C), implying that water-soluble AX promoted the thermal aggregation of gluten. However, these results do not seem to agree with previous studies that stated that AXs limited the effect of heat on gluten proteins [198,200]. P. Wang et al. [158] also reported that water-soluble AXs could promote the thermal aggregation of gluten. AXs also compete with gluten for water, hindering gluten hydration and delaying gluten network formation [178]. According to J. Li et al. [201], AXs with higher water absorption led to more significant water migration from the gluten network to the AXs, negatively affecting gluten network development [187], and, therefore, negatively affecting the quality of the final bread. There is a lot to be clarified and understood regarding the interaction of AXs with gluten and the importance on the final quality of the products. 

### 6.5. Beer

The impact of non-malted barley, rye, and oats on beer viscosity has been recently investigated by Langenaeken et al. [164]. They found that β-glucan content in barley and arabinoxylan content in rye results in a significant increase in beer viscosity compared to control beer. Furthermore, beer viscosity increase was linked to the arabinoxylan content and degree of polymerization. These experiments suggest that rye addition can increase the beer fullness of beers such as low-alcohol ones lacking in such sensory properties. Additionally, Li et al. [124] speculated that arabinoxylan may also have an important effect on the performance of foam stability on wheat beer.

To summarize, cereal AXs can exert functional roles in different and complex solid and liquid food systems due to their capacities for shelf-life extension, texturizing, bulking, and viscosifying. Additionally, because of their multifunctional attributes, AXs can be strategically included in a wide range of non-cereal food formulation such as infant-food (follow-on) formulas and non-alcoholic and pre/probiotic formulations with health-promoting features. 

## 7. Conclusions

This review aimed to gather the most recent information on AX’s structure, health benefits, and application for food products. From the information provided, it can be concluded that the heterogeneity of AXs is one of the main challenges that this dietary fibre presents to the research community. 

AX structure has a wide variability, being influenced by the source, state of germination, location within the grain, method of extraction and purification, and any further thermal, physical, chemical, or enzymatical treatments. Despite AX’s structural heterogeneity, different AXs present common potential health benefits, such as the modulation of SCFAs in the colon, improved antioxidant capacity, and reduced glucose blood response via different mechanisms. The available studies on these health benefits include in vivo and in vitro studies, usually with isolated AXs. However, the studies where AXs are integrated into food matrixes are limited. More studies about the health benefits of AXs integrated into food matrixes should be performed, because the food matrix could influence the observed health benefits, and not all AXs can be successfully included in food matrixes because of the poor physicochemical characteristics of enriched products. 

AXs have been shown to interact with other macromolecules present in food products, particularly those present in dough/bread. Modifying these AXs to deliver final products with acceptable characteristics could result in modified health effects. AX’s chemical structure has a determinant impact on the physical characteristics of food products. In general terms, AXs with lower Mw and high solubility seem to have better food applicability. However, the applied preprocessing procedures, extraction methods (parameters), and food processing technologies/parameters significantly affect AX’s properties and performance, indicating that AX’s functionality can be tailored to fulfill the desired end product qualities. The most widely studied product is bread, followed by pasta. Research on other food products regarding the inclusion of AXs is still limited.

## Figures and Tables

**Figure 1 foods-11-01026-f001:**
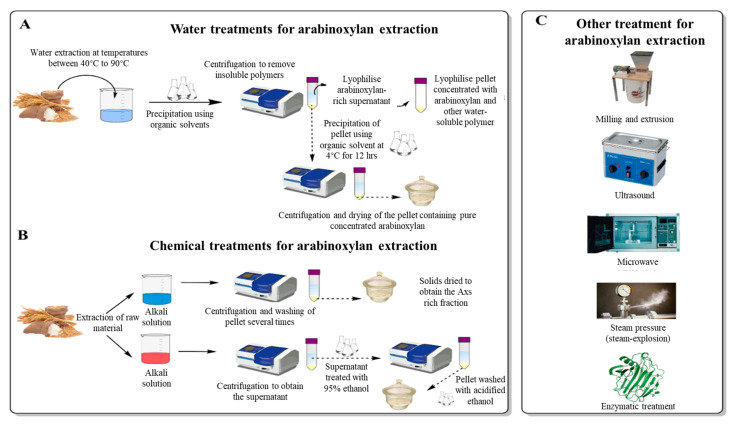
Schematic illustration of a water treatment approach (**A**) to extract AXs from cereal grains. (**B**) demonstrates a different approach using acidic or basic chemical solutions to extract AXs. Other treatments (**C**), including mechanical (milling and extrusion, steam-pressure, ultra-sound, microwave) and enzymatic treatments, are also included.

**Figure 2 foods-11-01026-f002:**
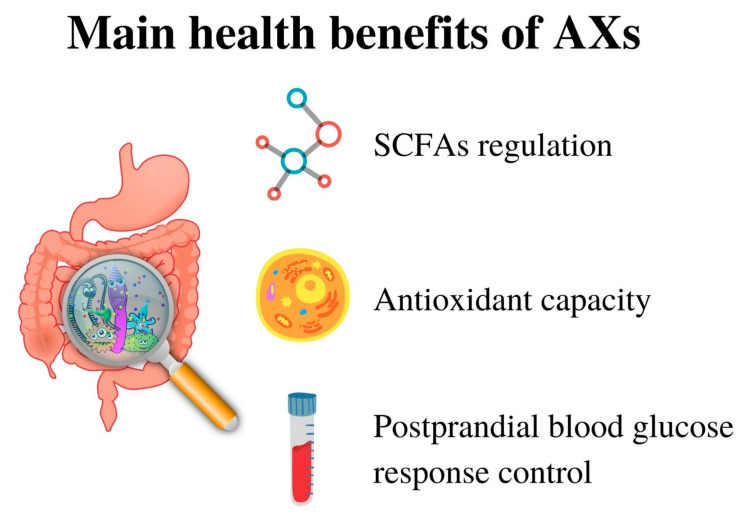
Main health benefits of arabinoxylans (AXs).

**Table 1 foods-11-01026-t001:** Summary of the main structural characteristics of total AXs and water-extractable AXs (WEAXs) found in different cereal grains. Xylan backbone substitutions between each AX differs. However, similar chemical structures are apparent amongst them. * The minor changes in these structural characteristics result in different interactive behaviours with other macromolecules.

Source of Arabinoxylan	Tissue Type	Total AXs (%)	WEAXs (%)	References	Main AX Structure *	References
**Wheat**	Endosperm	1.52–1.75	0.42–0.68	[14]	Side chains linked by α-(1→2) and/or α-(1→3) bonds along the xylan backbone. Xyloses are most commonly mono-substituted. Side chains formed mainly by single arabinose units but can contain other short sugar sidechains.	[12,15,16,17]
	Bran	11.0–16.4	0.54–0.95	[14]		
**Barley**	Endosperm	1.2–1.3	0.42–0.47	[18]	Similar structure to wheat AXs. Side chains of xylose units in the 2 and/or 3 carbon of the xyloses, which form the backbones of these AXs. Consists of more arabinose side chains than wheat AXs.	[19,20,21,22]
	Bran	10.26	-	[22]		
**Corn**	Cob	26.24	-	[23]	Highly branched structures with a xylose backbone. Side chains of arabinose residues on primary and secondary hydroxyl groups. Glucuronic acid, galactose, and xylose residues can also be present.	[24,25,26]
	Bran	26.0	0.71	[27]		
**Rice**	Endosperm	1.83	0.05	[28]	Characteristic sugar linkages and non-reducing end xylose and galactose. (1→2)-, (1→3)- or (1→5)-linked arabinose residues also present.	[29,30]
	Bran	6.82	011	[28]		
**Rye**	Endosperm	3.56–4.25		[31]	Main chain of 4-linked β-D-xylopyranosyl residues. A terminal α-L-arabinofuranosyl residue substitutes (on average) every second unit at position 3 and a small portion of the xylose units at position 2 and 3.	[32,33,34]
	Bran	12.6	2.1	[31]		
**Oat**	Endosperm	1.2	0.2	[35]	(1–4)-linked β-D-xylopyranosyl residues making up the main chain, with terminal L-arabinofuranosyl residues substituting at O-3, but also at both O-2 and O-3.	[35,36]
	Bran	5.2	0.7	[35]		

**Table 2 foods-11-01026-t002:** Various arabinoxylans extraction procedures and outcomes.

Source	Extraction	Solvent/Enzyme	AXs Yield *	A/X Ratio	Reference
De-starched wheat bran	Alkali	0.44 M NaOH	20.80	0.94	[83]
Corn fibre	Alkali	0.25–50 M NaOH	26.80 **	n.d.	[84]
De-starched plan materials	Alkali	NaOH (pH 11.5)	14.30–59.9 ***	n.d.	[57]
Chinese, black-grained wheat bran residue (after removal of water-extractable polysaccharides)	Alkali	Saturated Ba(OH)_2_, 1% NaBH_4_	~5.8	0.6	[85]
Wheat bran	Alkali	Saturated Ba(OH)_2_, 0.26 M NaBH_4_	24	0.7	[86]
Corn husk	Alkali	0.9% (*w*/*v*) Ca(OH)_2_	n.d.	0.75	[87]
De-starched wheat	Alkali/Enzymatic + alkali	0.16 mol/L NaOH, 0.5% H_2_O_2_//xylanase and cellulase (sodium acetate buffer) + 0.16 mol/L NaOH, 0.5% H_2_O_2_	19.83//5.27 and 14.95	1.14//0.25 and 1.52	[13]
Rye bran	Alkali + enzymatic	First extraction: 0.17 M Na_2_CO_3_ or 0.17 M Ca (OH)_2_ or water Second extraction: xylanase	First extraction: 2.92–3.85 Second extraction: 7.5–9.85	First extraction: 0.48–0.59 Second extraction: 0.23–0.28	[88]
Wheat and barley straw	Alkali and steam pretreatment + enzymatic	1–2 wt% NaOH (steam pretreatment) + β-glucosidase and xylanase	18–35 (Wheat) 17–47 (Barley)	n.d.	[89]
Wheat bran	Ultrasound + Enzymatic	Xylanase (sodium acetate buffer)	4.25–12.88	n.d.	[66]
Wheat bran	Enzymatic	Xylanase	23.1	0.44	[90]
Corn fibre	Enzymatic	Xylanase and cellulase (sodium acetate buffer)	30–45	n.d.	[90]

* AX extracted yield by raw material dry basis (% of Dw). ** Maximum yield achieved at optimized NaOH concentration, time, and temperature (0.5 M, 2 h, 60 °C). *** Yields were dependent on the material; yield could be influenced by pretreatments of these plant materials carried out by manufacturers. n.d.: not determined.

**Table 3 foods-11-01026-t003:** Most recent and relevant studies on the effects of arabinoxylans on the gut microbiome.

AXs Source	Type of Study	AXs Structure	Studied Parameters	Observed Effect	Reference
Triticale AXs extracted by different methods	In vitro	A/X ratio: 0.25–1.52	Ferulic acid content Antioxidant activity Hypoglycaemic activity	Esterified and free ferulic acid (FE) content was influenced by AX structure. Enzymatically or water-extracted AXs had higher levels of esterified FE, whereas alkali-extracted AXs had higher free FE levels. AXs with a lower degree of substitution contributed to higher antioxidant capacity. Alkali-extracted AXs performed better than other AXs in the inhibition of α-amylase, something that the authors correlated with the higher levels of free FE. Glucose absorption capacity by AXs was higher for enzymatically and water-extracted AXs, in contrast with the results of α-amylase inhibition.	[101]
Hard and soft wheat (whole grains)	In vitro-Human faecal fermentation	Water extractable AXs A/X ratio: 0.5 and 0.47 (soft and hard wheat, respectively) Mw: 410–4 kDa. Hard wheat had a higher % of AXs in the higher range.	Stimulation of *Bifidobacterium* and *Lactobacillus* growth SCFAs production Gas production	Significant stimulation of *Bifidobacterium* with AXs from hard wheat. Improvement of SCFAs contents: Increased acetic acid concentration (higher when using AXs from hard wheat). Increased propionic acid concentration. Increased butyric acid concentration (only for AXs from hard wheat and at the end of the fermentation process)	[102]
Hard and soft wheat (whole grains)	In vitro and in vivo (mice)	Water-extractable AXs A/X ratio: 0.5 and 0.47 (soft and hard wheat, respectively) Mw: 410–4 kDa. Hard wheat had a higher % of AXs in the higher range.	Relative growth of *Lactobacillus*, *Bifidobacterium*, *Bacteroides*, *Enterococcus,* and *Clostridium* Prebiotic activity SCFAs production	Increased growth and prebiotic activity of *Lactobacillus*, *Bifidobacterium*, and *Bacteroides* (only in vitro), decreased growth of *Clostridium*, no effect for *Enterococcus* (no data for in vitro) and *Bacteroides* (in vivo) for both AXs. Effects were higher with AXs from hard wheat. In vitro prebiotic activity was enhanced by AXs. In vivo results showed an improvement of SCFAs content (increased acetic acid and butyric acid (only with AXs from soft wheat) concentration)	[5]
Commercial corn bran AXs	In vivo (class-I obesity humans)	Long-chain AXs alkali extracted. A/X ratio: 0.56	Stool consistency and bowel movement frequency Faecal pH SCFA and moisture content Microbiota analysis	AXs altered global bacteria community and reduced bacterial diversity from week 1 of consumption, with no further changes with time. Bacterial shifts were highly individualised. AXs did not influence moisture content and faecal pH. AX consumption resulted in softer faecal consistencies and increased bowel movements. AXs did not modify total SCFAs concentration. AXs increased propionate relative abundance, and butyrate relative abundance decreased. Among the participants, two groups could be differentiated regarding propionate concentration along the intervention: group 1: concentration increased after one week but decreased after six weeks; group 2: concentration did not increase much after one week but sharply increased after six weeks. These two groups showed differences in microbiota between each other, although differences were not significant compared with the baseline.	[5]
Triticale bran	In vitro	Alkali-extractable AXs Hydrolysed alkali-extractable AXs Mw: AXs: 747 kDa AXs hydrolysed: 2.63–15.1 kDa. A/X ratio: 0.99 and 0.77–0.15 (hydrolysed AXs) Free and bound ferulic acid (FA and BA): FA > BA for all AXs.	Antioxidant activity Hypoglycaemic effect	Antioxidant capacity was increased when increasing AX concentration. For AXs with similar Mw, AX with a low degree of substitution (DS) had higher antioxidant activity. For AXs with similar DS, high Mw of AX was negatively correlated with its antioxidant activity at high DS. FA and BA in AXs were also important factors affecting its antioxidant activity. Hypoglycaemic effect: Positive, increased by AX concentration. Better for AXs with higher Mw (probably related to viscosity).	[107]
Corn bran (4 different genotypes)	In vitro-human faecal fermentation	Alkali extractable AXs A/X ratio: 0.46–0.54	SCFAs production Relative growth of bacteria	All AXs improved SCFAs production. Differences in SCFAs production (rate, abundance, and distribution) were related to corn genotypes. Different distributions of SCFAs among genotypes were correlated with the abundance of certain bacteria.	[101]

**Table 4 foods-11-01026-t004:** Main AX effect on food products.

Food Product	Main Observations	References or Patent Numbers
Pasta	Water-soluble AXs increase water absorption. AX structure is modified during elaboration process. AX loss drastically increases when cooking over the optimal cooking time. AX loss in the cooking water is higher when lower Mw of the AXs. The presence of added water-soluble AXs decreases pasta hardness.	[127,128,129,130]
Cookies	In general, AXs decrease spread ratio and increase hardness, although the effect is influenced by AX structure. AXs with very low Mw (oligosaccharides) increase the spread ratio. AXs of low Mw can be used to substitute sugar, increase the plasticity of the dough, and reduce the baking time.	[131,132,133,134,135,136,137]
Bread	AXs with high Mw have a detrimental effect on bread. AXs with ferulic acid increases dough extensibility. The combined use of AXs and enzymes can be an interesting strategy to increase specific volume and decrease staling. In addition, fermentation, extrusion, or sprouting of the flours/grains can also have a positive affect. AX modulates starch gelatinisation and retards the retrogradation of bread. AXs interfere with the gluten network. AXs with high Mw are more disruptive. Water-soluble AXs can increase bread volume, reduce rejuvenation, and extend bread’s shelf-life, inhibit the growth of ice crystal, and protect the dough network structure. Extending the shelf-life of bread also improves the flour performance.	[52,101,118,138,139,140,141,142,143,144,145,146,147,148,149,150,151,152,153,154,155,156,157,158,159,160,161,162,163]
		CN110938665A CN110938664A
Beer	AXs sourced from unmalted barley, rye, or oats improve the viscosity, fullness, and taste for low-alcohol beer. AXs promote the stability of wheat beer foam characteristics.	[124,164]
Cakes	β-glucan and arabinoxylans increase cake batter consistency and cell density and produce uniform crumbs while slowing down the movement of moisture from crumb to crust.	[165,166,167]
Infant formula milk powder	Promotes the growth and development of infants and toddlers.	CN108112702A
Infant and follow-on formulae	Controls the levels of glycemic index (Gl) and insulin index (II) in composite meal for infants and small children.	WO2015057151A1
Non-alcoholic beverages	Improves the mouthfeel of sugar and qualities of low calories beverages. Enhances the biological activities of the wheat-based drink. Lowers the glycemic responses on instant tea. Improves smoothness of oat-base beverages.	CN109843086A CN104522811A CN110897023A WO2014177304A1
Fish meal	Improves freezing resistance and nutritive value of fish ball.	CN112841568A

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
