# Peer review of "Arabinoxylans as Functional Food Ingredients: A Review"

_foods, 2022, doi:10.3390/foods11071026_

Round 1
Reviewer 1 Report
I reviewed the manuscript entitled, Arabinoxylans as functional food ingredients. A review. The review is up-to-date and addressed the review hypothesis; however, authors should consider few revisions.
Title can be revised as Arabinoxylans as functional food ingredients: A review
Abstract
The review objectives are clear but review findings and conclusions are not clear. This should be revised.
Lines 16 to 19: very long sentence
Keyword: format is not correct; just write keywords. No need of numbering
Section 2: graphical representation of AX structure in different sources should be provided. It will give clear idea on difference and/or similarities
Figure 1 is not readable. I suggest to increase the font size
Table 2. What is the unit of extraction yield?
Line 258: close the space after extracted Axs. Axs should be written as AXs
Table 3. In vitro or in vivo should be in Italics throughout the manuscript
Table 4. There are many products using AX. I suggest reviewing the latest studies dealing with incorporation of AX in food products.
Almost all the references are not according to journal format. Please revise
Author Response
Response to Reviewer 1 Comments
Point 1: Title can be revised as Arabinoxylans as functional food ingredients: A review
Response 1: The title has been modified as suggested by reviewer 1
Abstract
Point 2: The review objectives are clear but review findings and conclusions are not clear. This should be revised.
Response 2: The review findings and conclusions have been revised to improve their clarity and content (see lines 13-24).
Point 3: Lines 16 to 19: very long sentence
Response 3: The sentence has been re-drafted. (see lines 16-19)
Point 4: Keyword: format is not correct; just write keywords. No need of numbering
Response 4: The Keyword format has been adjusted following the journal guidelines
Point 5: Section 2: graphical representation of AX structure in different sources should be provided. It will give clear idea on difference and/or similarities
Response 5: The authors decide not to include the graphical representation of AX structure in different sources since such representation has already been included in several other reviews. Additionally, it would significantly increase the length of the manuscript.
Point 6: Figure 1 is not readable. I suggest to increase the font size
Response 6: The font size of the text in Fig 1 has been increased. (see line 157)
Point 7: Table 2. What is the unit of extraction yield?
Response 7: The unit of extraction yield has been added to the footnotes of table 2 (see line 265)
Point 8:Line 258: close the space after extracted Axs. Axs should be written as AXs
Response 7: The space after “extracted Axs” has been removed (see line 273)
Point 9: Table 3. In vitro or in vivo should be in Italics throughout the manuscript
Response 9. The “in vivo” and the “in vitro” format has been changed in italics throughout the manuscript.
Point 10: Table 4. There are many products using AX. I suggest reviewing the latest studies dealing with the incorporation of AX in food products.
Response 10: Latest studies have been included in table 4, and through the manuscripts.
Point 11: Almost all the references are not according to journal format. Please revise.
Response 1: A revision of the reference style has been performed.
Reviewer 2 Report
Dear Authors,
The work is too long and describes what is already well known in the field.
Chapters on extractions should be deleted or shortened.
40% of literature is over 10 years old.
The mauscript should be edited and supplemented with the latest news in this field, e.g.:
Effect of physicochemical properties, pre-processing, and extraction on the functionality of wheat bran arabinoxylans in breadmaking–a review
S Pietiäinen, A Moldin, A Ström, C Malmberg… - Food Chemistry, 2022 –
Food and Bioproducts Processing
Volume 132, March 2022, Pages 83-98
Food and Bioproducts Processing
Arabinoxylans: A new class of food ingredients arising from synergies with biorefining, and illustrating the nature of biorefinery engineering
Author links open overlay panelKonstantinaSolomouMohammadAlyassinAthanasiosAngelis-DimakisGrant M.Campbell
Isolation, Structural, Functional, and Bioactive Properties of Cereal Arabinoxylan─A Critical Review
Hong-Ju HeHong-Ju He
School of Food Science, Henan Institute of Science and Technology, Xinxiang 453003, China
More by Hong-Ju He, Jinli Qiao, Yan Liu, Qingbin Guo*, Xingqi Ou, and Xiaochan Wang
Cite this: J. Agric. Food Chem. 2021, 69, 51, 15437–15457.
Please format as required by the editorial staff.
I recommend major revision
Author Response
Response to Reviewer 2 Comments
The work is too long and describes what is already well known in the field.
Point 1: Chapters on extractions should be deleted or shortened.
Response 1: Chapter on Axs extractions has been shortened (see lines 133-276)
Point 2: 40% of literature is over 10 years old.
Response 2: The literature has been updated by including the latest publication focused on AXs
Point 3: The mauscript should be edited and supplemented with the latest news in this field, e.g.:
Effect of physicochemical properties, pre-processing, and extraction on the functionality of wheat bran arabinoxylans in breadmaking–a review
S Pietiäinen, A Moldin, A Ström, C Malmberg… - Food Chemistry, 2022 –
Food and Bioproducts Processing
Volume 132, March 2022, Pages 83-98
Food and Bioproducts Processing
Arabinoxylans: A new class of food ingredients arising from synergies with biorefining, and illustrating the nature of biorefinery engineering
Author links open overlay panelKonstantinaSolomouMohammadAlyassinAthanasiosAngelis-DimakisGrant M.Campbell
Isolation, Structural, Functional, and Bioactive Properties of Cereal Arabinoxylan─A Critical Review
Hong-Ju HeHong-Ju He
School of Food Science, Henan Institute of Science and Technology, Xinxiang 453003, China
More by Hong-Ju He, Jinli Qiao, Yan Liu, Qingbin Guo*, Xingqi Ou, and Xiaochan Wang
Cite this: J. Agric. Food Chem. 2021, 69, 51, 15437–15457.
Response 3: The manuscript has been updated by including the latest publication focused on Axs, including also those suggested by the reviewers
Point 4: Please format as required by the editorial staff.
Response 4: The manuscript has been format following the journal guidelines
I recommend major revision
Round 2
Reviewer 1 Report
Authors ignored to add few suggestions in revised manuscript. For example, Table 4. There are many products using AX. I suggest reviewing the latest studies dealing with the incorporation of AX in food products. For this, authors just added 1 line dealing with beer (only 1 reference. I exclusively suggest to review this section and add few more recent studies. Since, this is a review, it should at least cover more recent publications.
Author Response
Review’s comment
Point 1: Authors ignored to add few suggestions in revised manuscript. For example, Table 4. There are many products using AX. I suggest reviewing the latest studies dealing with the incorporation of AX in food products. For this, authors just added 1 line dealing with beer (only 1 reference. I exclusively suggest to review this section and add few more recent studies. Since, this is a review, it should at least cover more recent publications.
Reply point 1: The authors now include more studies and patents in table 4, which deals with the inclusion of AXs in different food and beverage products. However, studies on AXs as coating or antimicrobial films have not been included.
Reviewer 2 Report
Dear Authors,
The Authors responded to the comments. The manuscript may be published in present form.
Author Response
We are glad to read that the manuscript revision matches the reviewer's expectations.